# Carbutamide, an Obsolete Anti-Diabetic Drug, Has Potential as a Potent Anticolitic Agent via Azo-Conjugation with Mesalazine

**DOI:** 10.3390/pharmaceutics17121509

**Published:** 2025-11-22

**Authors:** Sanghyun Ju, Suji Kim, Taeyoung Kim, Jin-Wook Yoo, In-Soo Yoon, Eunsoo Kim, Yunjin Jung

**Affiliations:** 1College of Pharmacy, Pusan National University, Busan 46241, Republic of Korea; jsh141002@naver.com (S.J.); soozykim34@gmail.com (S.K.); taeyk@pusan.ac.kr (T.K.); jinwook@pusan.ac.kr (J.-W.Y.); insoo.yoon@pusan.ac.kr (I.-S.Y.); eunsoo_kim@pusan.ac.kr (E.K.); 2Research Institute for Drug Development, Pusan National University, Busan 46241, Republic of Korea

**Keywords:** carbutamide, colon-specific drug delivery, colitis, prodrug, drug repositioning

## Abstract

**Background:** To repurpose carbutamide (CBT), a discontinued sulfonylurea-class anti-diabetic drug, as an anti-inflammatory bowel disease (IBD) drug, CBT azo-linked with salicylic acid (CAA) was designed and synthesized as a colon-specific prodrug to co-release CBT and mesalazine (5-ASA) selectively in the large intestine. **Methods:** CAA exhibited reduced lipophilicity and decreased transintestinal transport compared to CBT, as shown in an ex vivo experiment using isolated rat jejunal segments. It also underwent cleavage into CBT and 5-ASA when incubated with cecal contents of rats. Additionally, oral administration of CAA and Sulfasalazine (SSZ), a colon-specific prodrug of 5-ASA currently used for IBD treatment, resulted in similar levels of 5-ASA accumulation in the rat cecal region. **Results:** In a dinitrobenzene sulfonic acid-triggered colitis model in rats, CAA produced a more pronounced improvement in colon injury and inflammation than SSZ. Furthermore, rectal co-administration of CBT and 5-ASA conferred enhanced protective outcomes compared to monotherapy with either agent alone, suggesting a combined anticolitic action. The two drugs also jointly suppressed valacyclovir uptake via peptide transporter 1 (PepT1) in the distal colon, supporting PepT1 as a target contributing to their combined anticolitic effect. Unlike CBT, which significantly reduced blood glucose following oral administration, equimolar administration of CAA did not alter glycemic levels, consistent with reduced systemic exposure to CBT. **Conclusions:** In conclusion, CAA functions as a colon-specific mutual prodrug that surpasses SSZ in anticolitic performance while minimizing hypoglycemia risk, thus facilitating the repurposing of CBT as a treatment for IBD.

## 1. Introduction

Inflammatory bowel disease (IBD) refers to a group of chronic inflammatory conditions that affect the gastrointestinal (GI) tract. The two predominant forms of IBD are ulcerative colitis and Crohn’s disease (CD). UC is typically localized to the innermost mucosal layer and predominantly occurs in the distal colon. In contrast, CD can involve multiple regions of the GI tract and is characterized by inflammation that penetrates deeper tissue layers, including the muscular layer [1,2]. Although extensive studies have been conducted over the past decades, the precise cause and pathogenesis of IBD remain incompletely understood. Individuals genetically susceptible to IBD are thought to possess immunological and physiological traits that trigger an aberrant immune response, leading to unrelenting inflammation when exposed to environmental stimuli that challenge immune homeostasis [2,3].

Conventional pharmacotherapy for IBD includes small-molecule drugs and biologics aimed at controlling inflammation and prolonging remission. Standard treatments such as aminosalicylates, corticosteroids, and immunosuppressants are often associated with limited anti-inflammatory efficacy or substantial adverse effects. Biologics, including anti-TNF-α agents, offer improved therapeutic outcomes but can still pose challenges related to safety, resistance, patient compliance, and medical cost [4,5]. More recently, small-molecule agents such as Janus kinase inhibitors and sphingosine-1-phosphate receptor modulators have been introduced, classified pharmacologically as immunosuppressants or immunomodulators. Despite these advances, issues like systemic toxicity, drug resistance, and parenteral administration continue to hamper long-term therapeutic success [6,7].

Drug repurposing (DR), also known as drug repositioning, is a strategy that seeks to identify new indications for existing or previously withdrawn drugs [8,9]. Given the high cost, failure rates, and extended timelines of de novo drug development, DR has garnered increasing interest, particularly for diseases lacking effective treatments. Repurposed drugs benefit from known pharmacokinetic profiles and established safety data, enabling faster and less expensive development. However, this approach is not without its challenges, including regulatory and medical hurdles [10]. In the case of DR for the treatment of IBD, colon-specific drug delivery has been explored as a means to improve the safety and efficacy of repurposed therapeutics [11,12].

Colon-specific drug delivery (CSDD) can be achieved through prodrug design or pharmaceutical formulations, allowing the active drug to bypass degradation and absorption in the upper GI tract and be released predominantly in the large intestine [13]. This site-specific delivery increases colonic drug concentration while minimizing systemic exposure, which enhances therapeutic effects and reduces side effects. CSDD is therefore particularly beneficial for diseases primarily affecting the large intestine [13,14]. For instance, mesalazine-based therapies are chemically modified or encapsulated to ensure delivery from the ileum to the distal colon [15]. Recent efforts have applied this strategy to DR, exploiting the advantages of CSDD to enhance efficacy and safety of repurposed drugs [11,16].

Sulfonylureas are orally active agents used in the management of type 2 diabetes mellitus. These compounds act on sulfonylurea receptors associated with ATP-sensitive potassium (KATP) channels, promoting insulin secretion by depolarizing pancreatic beta cells through calcium influx [17]. Notably, KATP channels are also expressed in monocytes and macrophages, where they modulate inflammatory responses via the MAPK and NF-κB signaling pathways. Accordingly, sulfonylureas have been reported to exhibit anti-inflammatory effects by attenuating cytokine release under both diabetic and non-diabetic conditions [18,19,20,21,22]. Furthermore, some sulfonylureas can inhibit inflammasome activation independently of KATP channel blockade by binding to sulfonylurea receptors [23,24]. Structural analysis suggests that the sulfonylurea moiety itself may underlie this activity, with variations in potency influenced by drug-specific chemical features [25]. 

Carbutamide (CBT), the first orally available anti-diabetic sulfonylurea, was originally derived from antimicrobial sulfonamides [26,27]. As a first-generation agent, CBT has been superseded by more modern sulfonylureas such as gliclazide, glipizide, glibenclamide, and glimepiride [26]. In addition to its potential anti-inflammatory effects, CBT and other sulfonylureas have been shown to inhibit peptide transporter 1 (PepT1), a transporter implicated in intestinal inflammation [28]. Inhibition of PepT1 has demonstrated efficacy in ameliorating experimental colitis by reducing uptake of pro-inflammatory peptides [29,30,31,32]. In this study, CBT-despite its discontinued clinical use due to cases of severe bone marrow and kidney toxicity reported in post-marketing surveillance—was selected for repurposing as an anticolitic agent using a colon-specific prodrug strategy [33]. This decision was based on three key factors: (1) CBT is expected to possess anti-inflammatory activity through inhibition of PepT1 in addition to inhibition of inflammasome and KATP channels, (2) CBT contains an aniline group suitable for azo-bond formation—a validated approach for colon-specific delivery [34], and (3) CBT is more hydrophilic than other sulfonylureas, which favors prodrug design for colonic delivery. Furthermore, major adverse effects associated with repurposing of CBT as an anti-IBD drug, including hypoglycemia and bone marrow suppression, may be mitigated through CSDD, potentially improving its safety profile. Notably, inhibition of inflammasome or PepT1 typically requires higher drug concentrations than KATP channel blockade, which are unlikely to be reached in plasma following standard oral dosing of sulfonylureas [25,28]. 

To this end, CBT was chemically linked to the anti-inflammatory agent mesalazine (5-ASA) via an azo bond, forming CBT azo-linked with salicylic acid (CAA) designed to deliver both agents to the large intestine. The colonic delivery performance of CAA was evaluated in parallel with sulfasalazine (SSZ), a clinically approved colon-specific 5-ASA prodrug [35]. While olsalazine, balsalazide, and sulfasalazine (SSZ) are all clinically relevant azo-prodrugs of 5-ASA, previous comparative studies have shown that their therapeutic efficacy is largely similar [35]. The therapeutic efficacy of CAA was assessed in a dinitrobenzene sulfonic acid (DNBS)-induced rat colitis model and benchmarked against SSZ. Additionally, potential synergy between CBT and 5-ASA was explored via intracolonic administration of the individual drugs and their combination. Finally, systemic safety was examined by comparing CBT plasma concentrations and hypoglycemic responses following oral administration of CBT versus CAA.

## 2. Materials and Methods

### 2.1. Materials

Salicylic acid and DNBS were sourced from Thermo Fisher Scientific Inc. (Waltham, MA, USA), while CBT was obtained from BLD Pharmatech Ltd. (Shanghai, China). 5-ASA, SSZ and Fluorescein isothiocyanate (FITC)-dextran (average molecular weight 4000) were supplied by Sigma-Aldrich Chemical Co. Inc. (St. Louis, MO, USA). Valacyclovir (VCV) was obtained from Ambeed, Inc. (Arlington Heights, IL, USA). Sodium nitrite (NaNO_2_) and sulfamic acid (HOSO_2_NH_2_) were purchased from Tokyo Kasei Kogyo Co. (Tokyo, Japan). Solvents used for reactions and those of HPLC grade were procured from Junsei Chemical Co. (Tokyo, Japan) and DAEJUNG Chemicals & Metals Co., Ltd. (Gyeonggi-do, Republic of Korea), respectively. The cytokine-induced neutrophil chemoattractant-3 (CINC-3) ELISA kit was obtained from R&D Systems (Minneapolis, MN, USA). Unless otherwise specified, all chemicals used in this study were of commercially available reagent grade. TLC analysis was performed using silica gel F254s plates (Merck Millipore, Burlington, MA, USA), and spot detection was carried out under UV light at 254 nm. High-resolution LC-MS/MS spectra were acquired using a ZenoTOF 7600 system (SCIEX, Framingham, MA, USA). Infrared spectra were recorded using a Varian FT-IR spectrophotometer, and proton nuclear magnetic resonance (^1^H-NMR) spectra were obtained with a Varian NMR instrument (Palo Alto, CA, USA). Chemical shifts in the ^1^H-NMR spectra are reported in parts per million (ppm) relative to tetramethylsilane as the internal standard.

### 2.2. Synthesis of CBT Azo-Linked with Salicylic Acid (CAA)

To CBT (271 mg) dissolved in pre-cooled 5 M HCl (15 mL), sodium nitrite (NaNO_2_, 103 mg) was gradually introduced under continuous stirring on ice for 30 min. Sulfamic acid (49 mg) was subsequently added to remove excess nitrite. In parallel, salicylic acid (138 mg) was solubilized in 1 M NaOH (5 mL), and the resulting solution was subsequently introduced into the diazonium mixture while maintaining the pH within 8–9 at a temperature of 20–25 °C for 6 h. Once the reaction was complete, the mixture was subjected to centrifugation at 3000× *g* for 10 min. The precipitate was rinsed three times with chilled distilled water and then dried under reduced pressure to obtain 5-((4-(N-(butylcarbamoyl)sulfamoyl)phenyl)diazenyl)-2-hydroxybenzoic acid (CBT azo-linked with salicylic acid, CAA) as reddish brown powder. Synthesis of CAA was verified using FT-IR, ^1^H-NMR spectroscopy, and time-of-flight mass spectrometry. CAA (M.W.: 420.44); Yield: 78%; mp: 186 °C (decomp.); FT-IR (nujol mull), ν_max_ (cm^−1^): 1672 (C=O, carboxylic and sulfonylurea, broad); ^1^H-NMR (500 MHz, DMSO-d_6_): *δ* = 8.39 (s, 1H), 8.11 (d, *J* = 9 Hz, 1H), 8.08 (d, *J* = 8.5 Hz, 2H), 8.03 (d, *J* = 8.5 Hz, 2H), 7.19 (d, *J* = 8.9 Hz, 1H), 2.95 (m, 2H), 1.31 (p, *J* = 7 Hz, 2H), 1.18 (sx, *J* = 7.3 Hz, 2H), 0.80 (t, *J* = 7.3, 3H); [M+*H*]^+^: *m/z* = 421.1177. 

### 2.3. HPLC Analysis

The HPLC system consisted of a Gilson model 306 pump, a 151 variable UV detector, and a model 234 autoinjector (Gilson, WI, USA). Chromatographic separation was carried out using a Symmetry C_18_ column (250 mm × 4.6 mm, 5 μm; VDS Optilab Chromatographietechnik GmbH, Berlin, Germany). Prior to HPLC injection, all samples were passed through 0.45 μm membrane filters. The mobile phase A was composed of distilled water and acetonitrile in a 6:4 volume ratio, while mobile phase B consisted of 1.0 mM phosphate buffer (pH 7.0), supplemented with 0.5 mM tetrabutylammonium chloride, and mixed with ACN in an 8.5:1.5 volume ratio (*v/v*). The chromatographic run was conducted at a fixed flow rate of 1 mL/min. Ultraviolet detection was carried out at wavelengths of 270 nm for CBT and 300 nm for 5-ASA, with the sensitivity adjusted to AUFS 0.01. Under these analytical settings, retention times were recorded as 6.5 min for CBT (in mobile phase A) and 9.5 min for 5-ASA (in mobile phase B).

### 2.4. Distribution Coefficient

Following the dissolution of CBT and CAA (1.0 mM) in 10.0 mL of 1-octanol pre-equilibrated with pH 6.8 phosphate buffer saline (PBS), an equal volume of the buffer (10.0 mL), which had been pre-saturated with 1-octanol, was layered onto the organic phase. The resulting biphasic system was gently agitated on an orbital shaker at 200 rpm for 12 h and then allowed to stand at 25 °C for 4 h to ensure complete phase separation. Absorbance at 270 nm was measured to determine the amount of each compound remaining in the 1-octanol layer using a UV-Vis spectrophotometer (Shimadzu, Kyoto, Japan). The distribution coefficient (log *D*_6__.__8_) was derived from the initial concentration in the organic phase (C_O_), the equilibrium concentration in the aqueous phase (C_W_), and the concentration in the 1-octanol phase at equilibrium (C_Oc_).$$logD_{6.8}=\log(C_{Oc}/C_{W})=log[{(C}_{Oc}/{(C}_{O}-C_{Oc})]$$

### 2.5. Chemical Stability

To evaluate the chemical stability of CAA, the compound (0.1 mM) was incubated for 10 h in two different buffer systems: an HCl-NaCl buffer at pH 1.2 and PBS. The drug concentrations over time were analyzed by HPLC to monitor potential degradation.

### 2.6. Animals

Seven-week-old male Sprague-Dawley (SpD) rats (Samtako Bio Korea, Gyeonggi-do, Republic of Korea) were maintained at the animal facility of Pusan National University (Busan, Republic of Korea) under standardized environmental conditions, including regulated temperature, humidity, and a 12-h light/dark cycle. All experimental procedures involving animals were reviewed and approved by the Institutional Animal Care and Use Committee of Pusan National University (PNU-IACUC), in accordance with ethical guidelines and scientific care standards (Approval No: PNU-2021-2942; Approval data: 23 March 2021).

### 2.7. Drug Release in Rat Small Intestinal and Cecal Contents

Male SpD rats were killed via carbon dioxide asphyxiation. Following euthanasia, a midline incision was performed on the abdomen to access the GI tract. The luminal contents of the small intestine and cecum were individually extracted and suspended in PBS to create a 20% (*w/v*) suspension. To maintain anaerobic conditions, cecal samples were processed inside a nitrogen-filled bag (AtmosBag, Sigma-Aldrich, MO, USA). A 2.0 mM solution of CAA, prepared in 3.0 mL of PBS, was mixed in equal parts (3.0 mL) with either intestinal or cecal suspensions and incubated at 37 °C. For cecal samples, the incubation was conducted under nitrogen atmosphere. At pre-determined time points, 0.5 mL of the mixture was withdrawn and centrifuged at 10,000× *g* for 10 min at 4 °C. From the resulting supernatant, 0.1 mL was transferred to 0.9 mL of MeOH, mixed using a vortex mixer, and centrifuged again under the same conditions. The final supernatants were filtered through 0.45 μm pore-size membrane filters, and CBT concentrations were quantified using HPLC.

### 2.8. Transport Assay Using Everted Gut Sacs of Rats

Male SpD rats underwent a 24 h fasting period, during which water was available without restriction. Rats were killed using carbon dioxide asphyxiation, and the jejunum (6 cm) were excised immediately. The jejunal was used for transport assay of CBT and CAA at 1.0 mM, respectively. The intestinal segments were well-rinsed with pre-warmed (37 °C) Krebs solution (120.0 mM NaCl, 5.0 mM KCl, 2.0 mM CaCl_2_, 1.0 mM MgCl_2_, 5.5 mM HEPES, and 1.0 mM d-glucose) adjusted to pH 7.4 and filtered through a 0.45 μm membrane filter for sterilization. The intestinal segments were gently everted using glass rods and forceps, and the apical surface was washed with the pre-warmed Krebs solution. One end of each segment was tied with thread, and the sac was filled with pre-warmed Krebs solution (600 μL), which was placed into a 15 mL tube containing pre-warmed Krebs solution (8 mL), ensuring that the sac was submerged to the solution so that the level of the internal solution became same with that of external solution. After securing 50 μL of the internal solution 10 min later as a blank used for background correction in UV absorption analysis, based on a preliminary experiment as described in Appendix A, the opposite end of the tube was loosely tied with thread and affixed to the outer surface of the tubing to hold the sac in place during the ex vivo studies. The assembled tubes were maintained at 37 °C, and aliquots (50 µL) were withdrawn from the basolateral (inner) side at scheduled time points, with an equal volume of fresh Krebs solution replenished each time. For quantitative analysis of CBT and CAA in the transport assay using jejunal segments, the samples and the blank obtained were mixed with MeOH (575 μL), centrifuged at 10,000× *g* at 4 °C for 10 min. The concentrations of CBT and CAA present in the supernatants were quantified at 270 nm (for CBT) and 355 nm (for CAA) using a UV-Vis spectrophotometer (Shimadzu, Kyoto, Japan). 

For the assessment of intestinal barrier integrity, an additional assay was conducted using FITC-dextran (100 μM) co-applied with either CAA (1 mM) or CBT (1 mM) to the apical side under the same experimental conditions as described in Appendix A.

### 2.9. Competition Study Using Everted Gut Sacs of Rats

Male SpD rats underwent a 24 h fasting period, during which water was available without restriction. Rats were killed using carbon dioxide asphyxiation, and distal colon (6 cm) were excised immediately. The colonic segments prepared and set as in Section 2.8 were used for competition study for transport via pepT1 using the pepT1 substrate VCV (0.5 mM) in the presence of 5-ASA or/and CBT (each at 5.0 mM). For quantitative analysis of VCV in the competitive study, the samples obtained from the basolateral side (50 µL) at scheduled time points were subjected to ethylacetate (EA) extraction to remove CBT, which interfered with the fluorescence analysis of VCV, and the aqueous layers isolated (25 µL) were mixed with MeOH (545 µL), centrifuged at 10,000× *g* at 4 °C for 10 min. VCV concentrations in the collected supernatant samples were determined at 280 nm (excitation)/370 nm (emission) using a fluorescence spectrophotometer (RF-6000, Shimadzu, Kyoto, Japan), following acidification of the supernatant (500 µL) with 1 M HCl (25 µL) [36]. FITC-dextran transport assay was conducted in the presence of CBT, 5-ASA, and CBT + 5-ASA under the same experimental conditions to confirm that the treatments did not affect epithelial barrier integrity as described in Appendix A.

### 2.10. Determination of Drug Concentration in Blood and Cecum 

Male SpD rats were subjected to a 24 h fasting period, during which they had free access to water. Each rat received one of the following oral treatments via gavage: CBT (20.4 mg/kg, corresponding to 31.5 mg/kg of CAA), CAA (31.5 mg/kg, equimolar to 30.0 mg/kg of SSZ), or SSZ (30.0 mg/kg). All compounds were suspended in 1.0 mL of PBS prior to administration. At 2, 5, and 8 h post-dosing, whole blood was obtained through cardiac puncture. Plasma was isolated by centrifuging the samples at 10,000× *g* for 10 min at 4 °C. To measure the plasma concentrations of CBT, 0.1 mL of plasma was combined with 0.9 mL of MeOH, vortex-mixed briefly, and centrifuged again under the same conditions. The supernatants were filtered through 0.45 μm membrane filters, and 20.0 μL aliquots were injected into the HPLC system to quantify concentrations of CBT. For analysis of drugs’ distribution in the cecum, SSZ (30.0 mg/kg) and CAA (31.5 mg/kg) were orally administered as PBS suspensions. At 2, 5, and 8 h post-administration, cecal contents were excised and suspended in PBS to create 10% (*w/v*) homogenates. After centrifugation (10,000× *g*, 10 min, 4 °C), 0.1 mL of the supernatant was mixed with 0.9 mL of MeOH, followed by vortexing. The solution was filtered through a 0.45 μm membrane filters, and aliquots (20.0 μL) were injected into the HPLC system to quantify concentrations of CBT and 5-ASA.

### 2.11. Determination of Blood Glucose Concentrations

Male SpD rats underwent an 18 h fasting period with unrestricted access to water. After receiving CBT (122.4 mg/kg) or its equimolar dose of CAA (190.5 mg/kg), each dissolved in 1.0 mL of PBS, blood glucose concentrations were assessed at appropriate time intervals. For this assessment, blood samples were collected from the murine tail vein. Briefly, anesthetized rats were firmly restrained to ensure clear visualization of the distal tail vein. The tail was thoroughly disinfected with alcohol, and the vein was punctured directly with a syringe needle, which was inserted in an upward direction, following the vessel’s path. The syringe plunger was slowly retracted to prevent venous collapse. After obtaining the desired volume of blood, the needle was removed, and the puncture site was compressed with gauze until hemostasis is achieved. The animals were then returned to their cages and carefully monitored for any signs of distress. Glucose levels were determined using the CareSens N Premier monitoring device (i-Sens Inc., Seoul, Republic of Korea), following the protocol recommended by the manufacturer.

### 2.12. DNBS-Induced Rat Colitis

Experimental colitis was established in rats following a previously reported method [37]. In short, male SpD rats were subjected to 24 h fasting with unrestricted access to water. Anesthesia was induced using isoflurane (Hana Pharm, Hwaseong, Republic of Korea) delivered through the Small Animal O_2_ Single Flow Anesthesia System (LMS, Pyeongtaek, Republic of Korea), with concentrations set at 3% for induction and maintained at 2.5% using 1 L/min of oxygen. Once anesthesia was confirmed by lack of response to tactile stimulation, a 2 mm outer-diameter rubber cannula was gently inserted into the rectum, positioning the tip approximately 8 cm from the anus at the level of the splenic flexure. A solution of DNBS (48.0 mg in 0.4 mL of 50% ethanol in water) was then administered through the cannula.

### 2.13. Evaluation of Anticolitic Effects

Two separate animal experiments were conducted to evaluate the therapeutic effects of various treatments against colitis. Treatment groups for the two animal experiments are shown in Appendix A. Three days after the induction of inflammation, colitic rats were medicated with each drug once per day via oral or rectal route, and the anticolitic effects were evaluated 24 h after receiving the medication. Colonic damage scores (CDS) were assigned based on a modified scoring system, in which 0 indicated normal tissue appearance; 1, localized hyperemia without ulceration; 2, linear ulcers with minimal inflammation; 3, a 2–4 cm region of ulceration and inflammation; 4, the same extent of inflammation with serosal adhesions to adjacent organs; and 5, the presence of strictures, multiple serosal adhesions involving several intestinal loops, and diffuse inflammation over 4 cm. For evaluation of myeloperoxidase (MPO) activity in inflamed colonic tissue, distal colon segments (4 cm) were finely chopped and transferred to tubes containing 1 mL of hexadecyltrimethylammonium bromide (HTAB) buffer (0.5% HTAB in 0.05 M phosphate buffer, pH 6.0). The samples were homogenized on ice using a T 10 basic ULTRA-TURRAX^®^ homogenizer (IKA Werke GmbH & Co. KG, Staufen im Breisgau, Germany). The homogenizer was rinsed with additional HTAB solution, and the pooled homogenate was diluted to a final concentration of 100 mg tissue/mL. The mixture was then centrifuged at 10,000× *g* at 4 °C. A 0.1 mL aliquot of the resulting supernatant was mixed with 2.9 mL of 0.1 M phosphate buffer (pH 6.0) containing o-dianisidine (0.3 mg/mL) and hydrogen peroxide (0.01%). The absorbance change at 460 nm was recorded for 5 min at 25 °C using a UV spectrophotometer (Shimadzu, Kyoto, Japan). One unit of MPO activity was defined as the amount of enzyme that catalyzes the degradation of 1.0 μmol of hydrogen peroxide per minute at 25 °C.

### 2.14. ELISA for CINC-3

Quantification of the pro-inflammatory chemokine CINC-3 in inflamed distal colonic tissue was carried out using enzyme-linked immunosorbent assay (ELISA) kits. Tissue samples from the distal colon were finely chopped in potassium phosphate buffer (pH 6.0), mechanically homogenized, and subsequently subjected to centrifugation at 10,000× *g* for 10 min at 4 °C. ELISA procedures were conducted in accordance with the protocols provided by the manufacturer.

### 2.15. Data Analysis

Data are presented as mean values accompanied by standard deviations (SD). Statistical comparisons among multiple groups were performed using one-way analysis of variance (ANOVA), followed by Tukey’s post hoc test. For CDS analysis, the non-parametric Mann–Whitney U test was employed. A *p*-value below 0.05 was considered indicative of statistical significance.

## 3. Results

### 3.1. Synthesis of CAA

As shown in Figure 1A, CAA was synthesized via azo coupling of CBT and salicylic acid. CBT dissolved in 5 M HCl was reacted with sodium nitrite to yield the diazonium salt of CBT followed by reaction with salicylic acid dissolved in an alkaline solution. CAA was precipitated when the reaction mixture was adjusted to pH 8–9. The synthesis proceeded smoothly and the reaction yield was over 70%. Formation of CAA was confirmed by FT-IR, ^1^H-NMR, and mass spectrometry. In IR spectrum (Appendix A) of CAA, the absorption bands for the carbonyl groups derived from the sulfonylurea in CBT and the carboxylic group in 5-ASA appeared as a broad one with a peak at 1672 cm^−1^. In ^1^H-NMR spectra (Appendix A) of CAA and CBT, the proton peaks derived from the aromatic ring of CBT, downfield-shifted by azo conjugation with salicylic acid, were detected at 8.03 and 8.08 ppm along with the proton peaks (8.39, 8.11, 7.19 ppm) derived from the aromatic ring of azo-conjugated salicylic acid. Aliphatic protons derived from CBT were observed without significant change in chemical shift. In addition, the proton peak of the aniline group in CBT, detected at 6.06 ppm, disappeared in the CAA spectrum, resulting from azo-conjugation. Finally, the molecular peak of CAA was clearly detected at 421.1177 [M + H], corresponding to molecular weight of CAA (Appendix A). The proposed colonic activation pathway of CAA to liberate CBT and 5-ASA is shown in Figure 1B.

### 3.2. CAA Is Colon-Specific

We first examined whether CAA was colon-specific. To do this, the intestinal transport of CAA and CBT was compared to assess whether azo conjugation with salicylic acid retarded passive transport across intestinal wall. Distribution coefficients of CBT and CAA, a physicochemical parameter predicting membrane transport, were determined in n-octanol/water system. At the same time, intestinal transport was measured using jejunal segments isolated from rats. The Distribution coefficient of CBT decreased from 1.23 to −0.47 upon azo-conjugation with salicylic acid, likely due to an increase in the number of polar groups. In parallel, azo-conjugation with salicylic acid substantially retarded the transport of CBT across the intestinal wall of the jejunal segments (Figure 2A), indicating less efficient transport of CAA through the intestinal wall including epithelial and mucosal layers. To verify that the transport was not affected by drug-induced epithelial barrier damage, intestinal barrier integrity was assessed using FITC–dextran co-applied with either CAA or CBT to the apical (donor) side under the same experimental conditions. The transport of FITC–dextran was very low (the basolateral concentration of FITC-dextran was approximately 0.5% of the apical concentration of FITC-dextran at 90 min) and showed no differences among the control, CAA-, and CBT-treated groups, indicating no drug-induced damage to the intestinal barrier. In addition, the mass loss of CBT and CAA was less than 1.5 and 1.4% of the total amount of drugs, respectively, during the transport assay (Appendix A). These results suggest that systemic absorption of CAA can be limited during transit of the GI tract, likely allowing a substantial portion of orally given CAA to reach the large intestine. In addition, a colon-specific prodrug should be activated to its parent drug within the large intestine. To test whether CAA satisfied the colon-specific condition, CAA was incubated with cecal contents under nitrogen, simulating anaerobic condition in the large intestine, and its conversion to CBT was monitored. As shown in Figure 2B, CAA was converted to CBT, whose conversion percent was 65.6% at 2 h and 77.4% at 8 h. When the same experiment was conducted in the cecal contents autoclaved at 121 °C for 15 min [38], no significant change in the concentration of CAA was observed during 24 h incubation, indicating requirement of microbial enzymes for the CAA activation. During incubation with a 10% suspension of small intestinal contents, the concentration of CAA remained stable for up to 10 h, indicating minimal degradation in the upper GI environment. To evaluate its colon-targeting capability, the amount of 5-ASA reaching the large intestine was compared between orally administered CAA and SSZ, a known colon-specific prodrug of 5-ASA. As depicted in Figure 2C, 5-ASA levels in the cecal region were similar at all measured time points following administration of either compound. Furthermore, CBT concentrations in the cecum were found to be similar to those of 5-ASA, as shown in Figure 2D. These findings collectively suggest that CAA selectively reaches the large intestine, where it undergoes bioconversion to release both CBT and 5-ASA.

### 3.3. CAA Is More Effective Against DNBS-Induced Rat Colitis than SSZ

To evaluate whether CAA could alleviate colitis and whether its metabolites, 5-ASA and CBT, exert a combined therapeutic effect in the large intestine, we conducted in vivo testing in a DNBS-induced rat colitis model. CAA was orally administered once daily for 6 days to rats subjected to colonic inflammation via rectal DNBS administration, and its anti-inflammatory efficacy was assessed 24 h after the sixth medication. A parallel group received SSZ, known to release 5-ASA in the colon, under the same conditions. SSZ was used as a reference drug because other types of colon-specific prodrugs of 5-ASA, olsalazine and balsalazide, do not differ from SSZ in terms of efficacy and tolerability in UC [35,39]. Additionally, a mixture of CBT and 5-ASA was orally administered to investigate the importance of colon-specific release in mediating therapeutic action. As shown in Figure 3A,B, DNBS induced pronounced colonic injury, including ulceration, hemorrhagic crust formation, tissue swelling, lumen narrowing, and adhesions to adjacent organs. CAA markedly reduced these pathological features in a dose-responsive fashion and demonstrated superior therapeutic outcomes compared to an equimolar dose of SSZ. To further visualize tissue recovery, hematoxylin and eosin (H&E) staining was performed. As illustrated in Figure 3C, CAA significantly improved DNBS-induced mucosal damage, with greater restorative effects than SSZ at the same molar dose. In line with these histological findings, measurement of MPO activity—a marker of neutrophil infiltration-revealed a notable reduction in inflammation in the CAA-treated group. Elevated MPO levels seen in DNBS-only rats were substantially lowered by CAA treatment, exceeding the anti-inflammatory effect observed with SSZ (Figure 3D). Furthermore, quantification of the chemokine CINC-3 in the inflamed tissue showed that CAA suppressed this inflammatory mediator more effectively than SSZ, consistent with earlier observations (Figure 3E). Interestingly, administration of a CBT and 5-ASA mixture without colon-specific targeting yielded limited protective effects. These findings highlight that CAA’s ability to concurrently deliver CBT and 5-ASA specifically to the large intestine is key to its enhanced efficacy over SSZ. Thus, CAA can be considered a dual-action colon-specific prodrug with superior therapeutic potential for the treatment of colitis.

### 3.4. Rectal Co-Administration of CBT and 5-ASA Is More Effective Against DNBS-Induced Rat Colitis than Monotherapies with Each Drug

To verify whether CAA functions as a colon-specific mutual prodrug in treating colitis, CBT and/or 5-ASA were delivered via the rectal route to colitic rats once daily for 6 days. According to Figure 4A (colon morphology), Figure 4B (CDS), and Figure 4C (H&E-stained sections), co-administration of CBT and 5-ASA yielded greater recovery of colonic and mucosal damage compared to monotherapies with either agent alone. A similar trend was observed when neutrophil infiltration was evaluated via MPO activity (Figure 4D). The combination treatment significantly reduced MPO levels compared to individual treatments. This enhanced anti-inflammatory effect was further supported by decreased expression of CINC-3. As illustrated in Figure 4E, the combined administration led to a greater reduction in CINC-3 levels than treatments with CBT or 5-ASA alone. Rectally administered 5-ASA showed comparable efficacy in mitigating colonic inflammation and tissue injury to that of CBT. Collectively, these findings suggest that CBT localized at the inflammation site contributes to therapeutic effects and cooperates with 5-ASA, supporting the conclusion that CAA operates as a colon-specific mutual prodrug for treating colitis in rats.

### 3.5. CBT and 5-ASA Cooperate to Inhibit Transport of VCV via PepT1 

PepT1, which is upregulated in the colons of patients with IBD [40], represents a promising therapeutic target for anti-IBD drug development. Both sulfonylureas and 5-ASA have been reported to inhibit PepT1 activity [28,41]. To investigate whether the combined anticolitic activity shown by CBT and 5-ASA is linked to PepT1 inhibition, we evaluated the transport of VCV—a known PepT1 substrate [42], across the distal colonic wall of rats in the presence of CBT and/or 5-ASA. PepT1 is expressed in the distal colon of both rodents and humans [43]. As depicted in Figure 5, individual application of CBT or 5-ASA similarly reduced the translocation of VCV through the everted distal colon wall, implying that both agents serve as PepT1 inhibitors with comparable inhibitory strength. Furthermore, their combination resulted in a greater suppression of VCV transport than each single treatment. Intestinal barrier integrity was assessed using FITC-dextran co-applied with drugs to the apical (donor) sides under the same experimental conditions, and the transport of FITC-dextran remained very low (the basolateral concentration of FITC-dextran was approximately 0.5% of the apical concentration of FITC-dextran at 90 min) with no differences among the control, CBT-, 5-ASA-, or combination-treated groups, thereby ruling out the possibility that the observed change in VCV transport was due to drug-induced epithelial damage. In addition, the mass loss of VCV was less than 1% of the total amount of the drug during the competitive assay (Appendix A). These findings suggest that cooperative inhibition of PepT1 contributes to the combined therapeutic action observed with CBT and 5-ASA in colitis treatment.

### 3.6. CAA Reduces the Risk of Hypoglycemia of CBT by Limiting Its Systemic Absorption

Our findings suggest that CAA may offer therapeutic advantages over SSZ. SSZ, an azo linkage connecting 5-ASA to sulfapyridine, is associated with systemic adverse effects due to the release of sulfapyridine, leading to treatment discontinuation in a notable proportion of IBD patients [4]. Given that CBT’s original pharmacological activity—hypoglycemia—could pose unintended clinical concerns for non-diabetic individuals when repurposed as an anti-IBD agent, it was essential to evaluate whether colonic delivery of CBT via CAA could minimize such risks. Targeted drug release to the large intestine is generally considered to limit systemic exposure, thereby lowering the incidence of unwanted systemic effects [13]. To assess this, both CBT and CAA were orally administered to rats, and plasma levels of CBT were measured at 2, 5, and 8 h following administration. As depicted in Figure 6A, orally administered CBT resulted in blood levels of CBT ranging from 72.6 to 96.8 μM across the measured time points, with the maximum value recorded at 5 h. In contrast, CBT derived from orally administered CAA exhibited consistently lower plasma levels throughout the observation period, reaching no more than half of the peak value seen with CBT intake. To confirm whether this limited systemic exposure correlates with a reduced likelihood of hypoglycemic side effects, we designed an additional experiment. Since a 20.4 mg/kg dose of CBT, used for its colitis-ameliorating action, did not cause significant reductions in blood glucose (Appendix A), the dose was escalated to 122.4 mg/kg to produce a measurable hypoglycemic response. This allowed us to compare the systemic safety profiles of CBT and CAA under conditions where CBT demonstrates noticeable pharmacological effects. Guided by the pharmacokinetic profile of CBT, which showed peak plasma levels at 5 h post-administration, blood glucose was measured at this time point. As shown in Appendix A, high-dose CBT led to a marked decrease in glucose levels, reaching approximately 60% of baseline. Subsequently, blood glucose concentrations were monitored at multiple intervals following oral administration of either CAA (at a CBT-equivalent dose) or high-dose CBT. According to Figure 6B, blood glucose in the CBT group dropped to 61% at 5 h and remained below the normal level up to 24 h, despite a gradual recovery. In contrast, CAA-treated rats maintained normal and stable glucose levels throughout the entire monitoring period. Collectively, these observations demonstrate that, beyond its therapeutic efficacy, colonic targeting of CBT via CAA is anticipated to improve its safety profile by mitigating systemic side effects such as hypoglycemia when repurposed for IBD treatment.

## 4. Discussion

CBT, though no longer in clinical use, is an orally active first-generation sulfonylurea anti-diabetic agent [17,27]. Based on accumulating evidence that sulfonylureas exhibit anti-inflammatory properties [19], CBT was considered a candidate for therapeutic repurposing as an anti-inflammatory agent for the treatment of IBD. To facilitate this repurposing, a colon-specific prodrug strategy was adopted. 

A colon-specific derivative of CBT was synthesized by linking it to salicylic acid through an azo bond, forming CBT azo-linked with salicylic acid (CAA). This azo linkage, commonly used in prodrug design for compounds containing an aniline structure like CBT, is well established for its selective activation in the colon [13,44]. This structural modification enables CAA to release both CBT and 5-ASA specifically in the large intestine through microbial reduction of the azo bond by colonic azo reductases [13]. Our findings confirmed that the azo bond in CAA remained intact in the small intestine and acidic conditions, but was effectively cleaved in the cecal environment, fulfilling a key requirement for colon-selective activation [13]. Additionally, azo linkage with SA significantly increased the hydrophilicity of CBT, as reflected by a lower log D_6__.__8_ value for CAA. This enhanced polarity, attributed to salicylic acid’s hydroxyl and carboxyl groups, likely contributed to the reduced passive transport of CAA across the intestinal wall, as demonstrated by ex vivo transport studies using the jejunum isolated from rats. These physicochemical and transport characteristics align with our in vivo data, which showed that CAA reached the cecum intact and efficiently released 5-ASA at levels comparable to those achieved by SSZ, a clinically used colon-specific prodrug of 5-ASA [4]. 

The design of this colon-specific prodrug was intended not only to enhance the local activity of CBT in the colon, but also to promote therapeutic cooperation with 5-ASA in suppressing colitis. Supporting this concept, our in vivo results demonstrated that orally administered CAA alleviated colonic inflammation and tissue injury more effectively than SSZ or a physical mixture of CBT and 5-ASA. This suggests that the therapeutic advantage of CAA over SSZ stems from cooperative interplay between its two active components specifically delivered to the target site. The cooperative action was further supported by rectal co-administration studies, where the combination of CBT and 5-ASA showed greater efficacy in reducing rat colitis than monotherapies with either compound alone. 

Interestingly, when orally given as a simple mixture, CBT and 5-ASA failed to show significant protective effects, implying that the CAA-mediated therapeutic enhancement is closely related to local action resulting from site-specific release of CBT and 5-ASA within the colon. Therefore, we propose that the combined therapeutic efficacy may be partially mediated by cooperative inhibition of PepT1, a transporter that is overexpressed in the colonic epithelium of IBD patients and believed to facilitate uptake of bacterial pro-inflammatory peptides [29,32,45]. Our study found that both CBT and 5-ASA suppressed transport of VCV, a substrate of PepT1, across the distal colon, and their combined use resulted in a stronger inhibition than either compound alone [42]. Notably, CBT’s inhibitory capacity was comparable to that of 5-ASA, aligning with previous findings on 5-ASA’s activity [41]. Although the precise mechanism of their cooperative therapeutic action requires further clarification, the observed additive inhibition of PepT1 supports its involvement in their combined efficacy against colitis. In addition, these result may be clinically relevant, given that the presence of PepT1 mRNA and protein expression in the distal colonic epithelium of rats and humans and PepT1 protein upregulated in colonic epithelia of inflammatory bowel disease patients [43]. 

Beyond its therapeutic benefit, CAA also offers a safety advantage by potentially minimizing systemic exposure to CBT. This was demonstrated by pharmacokinetic analysis, where plasma CBT concentrations were significantly lower following oral CAA administration than with direct CBT administration, with peak levels reduced by approximately half. The delayed systemic appearance of CBT (as shown in Figure 6A) after oral CAA administration suggests that absorption primarily occurred in the colon, facilitated by the time required for CAA to transit to the lower intestine before drug release. Given the large intestine’s relatively low fluid content, reduced motility, and wider lumen—factors that impede efficient drug absorption—CBT liberated from CAA was absorbed to a lesser extent than orally administered CBT, which was primarily absorbed in the small intestine. Importantly, while CBT was still detectable in circulation after oral CAA administration, it did not induce the hypoglycemic effect observed with oral CBT administration at an equimolar dose, highlighting the attenuation in systemic effects via CSDD. These findings underscore CAA’s potential to mitigate adverse effects such as hypoglycemia and bone marrow toxicity [33] upon drug repositioning of CBT to an anti-IBD drug. 

Although the DNBS-induced colitis model is widely used to study intestinal inflammation, it has inherent limitations in recapitulating the complex pathophysiology of human inflammatory bowel disease (IBD) [46]. DNBS-induced colitis primarily represents an acute, T cell-mediated immune response, whereas human IBD is characterized by a chronic and relapsing course involving multifactorial immune dysregulation, microbial interactions, and genetic susceptibility. Furthermore, the DNBS model lacks the genetic and environmental heterogeneity seen in patients and does not fully reproduce the epithelial barrier dysfunction and microbiome alterations that contribute to human disease progression [47]. Therefore, our findings should be interpreted with caution, and extrapolation to clinical settings should be made carefully, acknowledging these translational limitations. 

Historical reports have documented that high systemic doses of CBT may cause bone marrow suppression and kidney toxicity, raising concerns about its safety profile [48]. In this study, our colon-targeted prodrug approach is designed to deliver CBT locally to the colon, which is expected to significantly reduce systemic exposure and consequently minimize these toxicities. Nevertheless, the potential for bone marrow and other systemic toxicities cannot be excluded, especially with long-term treatment. Therefore, future preclinical studies are essential to systematically evaluate these safety concerns. 

In this study, we primarily focused on the functional inhibition of PepT1-mediated substrate transport rather than direct modulation of PepT1 protein expression in colonic tissue. While molecular or immunohistochemical analyses could provide additional mechanistic insights, to date, no small molecules have been reported to reduce PepT1 protein levels in normal colonic tissue. Future investigations incorporating molecular and immunohistochemical approaches will be essential to elucidate whether the tested compounds also influence PepT1 mRNA and protein expression in the colon.

In conclusion, CAA, produced by azo conjugation of CBT with salicylic acid, acts as a colon-specific mutual prodrug with superior efficacy to SSZ against rat colitis and reduces the risk of systemic side effects of CBT. Therefore, CAA may provide a promising therapeutic option for the treatment of IBD substituted for SSZ.

## Figures and Tables

**Figure 1 pharmaceutics-17-01509-f001:**
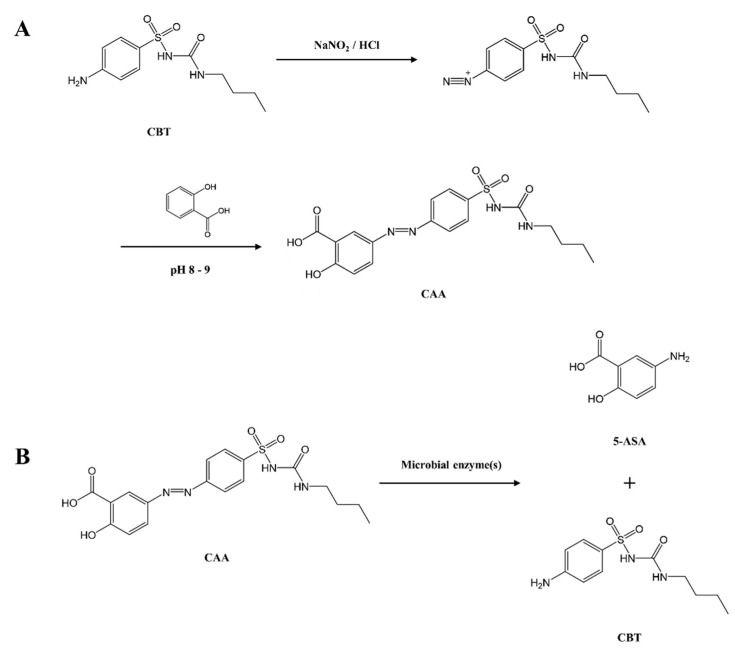
Synthesis and colonic activation of CAA. (**A**) Synthetic scheme of CAA. (**B**) Proposed metabolic pathway involving microbial activation of CAA into CBT and 5-ASA. CBT: carbutamide, CAA: CBT azo-linked with salicylic acid, 5-ASA: Mesalazine.

**Figure 2 pharmaceutics-17-01509-f002:**
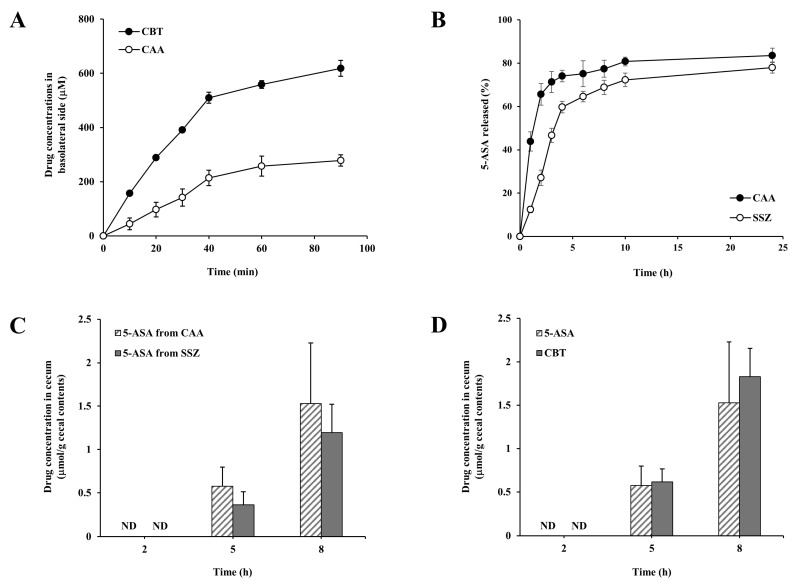
CAA exhibits colon-specific properties. (**A**) The everted jejunal segments were submerged to the pre-warmed Krebs solution (8 mL) where CBT or CAA (each at 1.0 mM) was dissolved in a 15 mL tube, and incubated for 90 min. To maintain osmotic balance, the fluid levels inside and outside the gut sacs were matched. During this period, samples (50 µL) were collected from the basolateral (inner) side filled with Krebs solution (600 µL) and replaced with equal volumes of fresh Krebs solution. Drugs in the samples were quantified using a UV-Vis spectrophotometer. (**B**) CAA and SSZ were incubated in a 10% suspension of cecal contents in PBS and release of 5-ASA was analyzed using HPLC. (**C**) Rats received an oral dose of CAA (31.5 mg/kg, equimolar to 30.0 mg/kg of SSZ) or SSZ (30.0 mg/kg), suspended in PBS (1 mL), and were euthanized at 2, 5, and 8 h post-administration. Cecal concentrations of 5-ASA were determined by HPLC. (**D**) In a parallel experiment, concentrations of CBT and 5-ASA released from CAA were measured under the same conditions. Data are expressed as mean ± SD (*n* = 5).

**Figure 3 pharmaceutics-17-01509-f003:**
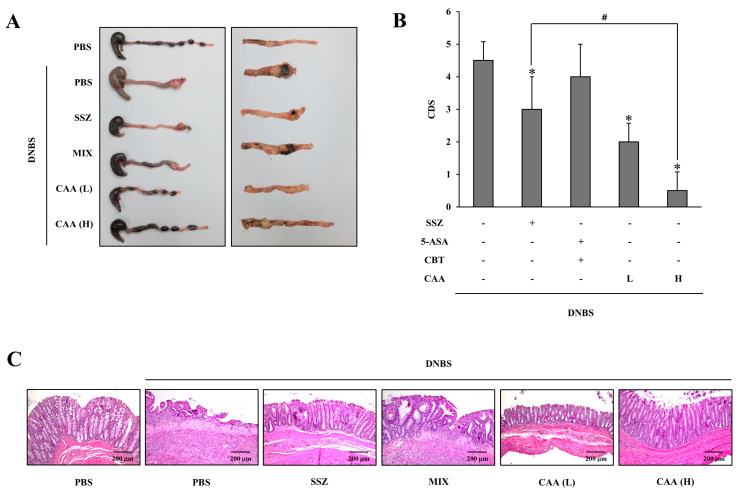
CAA demonstrates superior efficacy to SSZ in alleviating DNBS-induced rat colitis. Three days after induction of colitis with DNBS, rats received once-daily oral administration of either SSZ (30.0 mg/kg), a combination of CBT (20.4 mg/kg) and 5-ASA (11.5 mg/kg) (MIX), low-dose CAA (L, 15.8 mg/kg), or high-dose CAA (H, 31.5 mg/kg, equivalent to 20.4 mg/kg of CBT; molar equivalent to 30.0 mg/kg of SSZ), each suspended in 1.0 mL of PBS, for six consecutive days. The rats were euthanized 24 h after the final dose. (**A**) Representative images of the serosal and luminal sides of the distal colon are presented. (**B**) CDS was assessed across treatment groups. (**C**) Histological changes were evaluated by H&E staining of the inflamed distal colon. (**D**) MPO activity in the inflamed distal colon (4.0 cm) was quantified. (**E**) Levels of CINC-3 in the inflamed distal colon were measured by ELISA. Data are expressed as mean ± SD (*n* = 5). * *p* < 0.05 vs. DNBS control; ^#^ *p* < 0.05. The exact *p*-values are provided in the Appendix A.

**Figure 4 pharmaceutics-17-01509-f004:**
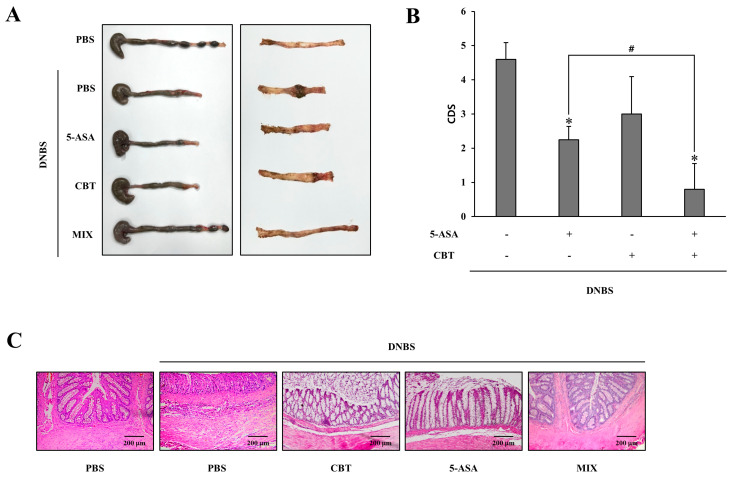
Rectal co-administration of CBT and 5-ASA is more effective against DNBS-induced rat colitis than monotherapies with each drug. Three days after induction of colitis with DNBS, rats received once-daily rectal administration of CBT (30 mM) and/or 5-ASA (30 mM) suspended in PBS (0.5 mL) for six consecutive days. MIX: CBT (30 mM) and 5-ASA (30 mM). The rats were euthanized 24 h after the final dose. (**A**) Representative images of the serosal and luminal sides of the distal colon are presented. (**B**) CDS were determined for each treatment group. (**C**) Histological evaluation was performed using H&E staining of colonic sections. (**D**) MPO activity in the inflamed distal colon (4 cm) was quantified. (**E**) Levels of CINC-3 in the inflamed distal colon were measured by ELISA. Data are expressed as mean ± SD (n = 5). * *p* < 0.05 vs. DNBS control; ^#^ *p* < 0.05. The exact *p*-values are provided in the Appendix A.

**Figure 5 pharmaceutics-17-01509-f005:**
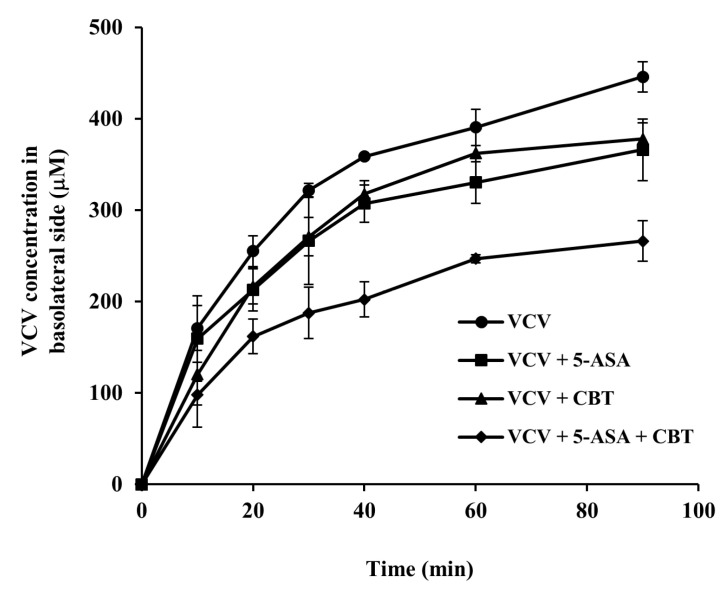
CBT and 5-ASA cooperate to inhibit PepT1-mediated transport of VCV in the distal colon isolated from rats. Rats were fasted for 24 h with free access to water, and segments of the distal colon (6 cm) were rapidly excised following euthanasia. The isolated and everted colonic tissues were mounted in a competitive transport assay using VCV (0.5 mM), a known PepT1 substrate, in the presence of CBT and/or 5-ASA (each at 5 mM). The everted gut sacs were filled with pre-warmed Krebs solution (600 µL) and immersed in 15 mL tubes containing Krebs solution (8 mL) pre-warmed at 37 °C. To maintain osmotic balance, the fluid levels inside and outside the gut sacs were matched. At predetermined time points, 50 µL aliquots were collected from the basolateral (inner) side and replaced with equal volumes of fresh Krebs solution. VCV concentrations were determined using a fluorescence spectrophotometer at 270 nm (excitation)/370 nm (emission) following removal of CBT by EA extraction. Data are expressed as mean ± SD (*n* = 5).

**Figure 6 pharmaceutics-17-01509-f006:**
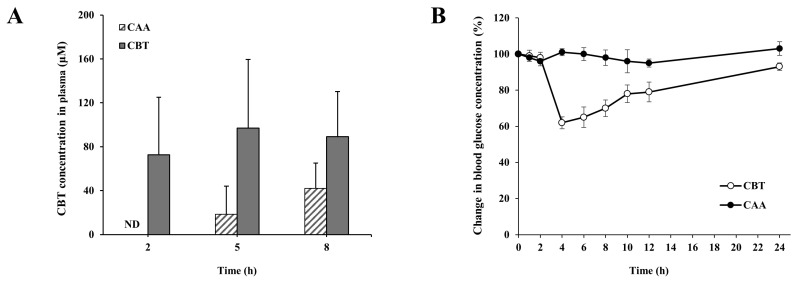
CAA reduces systemic exposure and hypoglycemic risk of CBT. (**A**) Rats were fasted for 24 h with free access to water, then given CBT (20.4 mg/kg) or CAA (31.5 mg/kg), suspended in PBS (1 mL). At 2, 5, and 8 h post-dose, the rats were euthanized and the blood samples were collected via cardiac puncture. Blood concentrations of CBT were quantified by HPLC. (**B**). In a separate experiment, rats fasted under identical conditions received CBT (122.4 mg/kg) or an equimolar dose of CAA (190.5 mg/kg), and blood glucose concentrations were monitored at multiple time points following oral administration. Data are expressed as mean ± SD (*n* = 5).

## Data Availability

The raw data supporting the conclusions of this article will be made available by the authors on request.

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
