# Peer review of "Carbutamide, an Obsolete Anti-Diabetic Drug, Has Potential as a Potent Anticolitic Agent via Azo-Conjugation with Mesalazine"

_pharmaceutics, 2025, doi:10.3390/pharmaceutics17121509_

Round 1
Reviewer 1 Report
Comments and Suggestions for Authors
The choice of carbutamide (CBT) is well-justified based on its anti-inflammatory potential, PepT1 inhibition, and chemical suitability for azo conjugation.
The "mutual prodrug" concept for drug repurposing in IBD is interesting. They designed carbutamide azo-linked with SA (CAA) as a colon-specific prodrug to co-release CBT and 5-ASA in the large intestine. The inclusion of both oral and rectal administration studies separates local vs. systemic effects.
-The first sentence of the abstract is too long. It is suggested to rewrite it and improve the readability of the abstract.
-It is suggested to report data on inflammatory cytokines (TNF-α, IL-1β, IL-6) to better elucidate the anti-inflammatory mechanisms.
-Performing the chronic toxicity assessment (as future work) would be valuable for a repurposed drug.
-It is suggested to include exact p-values in the results section in the manuscript and figure legends
-It is recommended to elaborate more on the potential limitations of the DNBS model for human IBD translation.
Reviewer 2 Report
Comments and Suggestions for Authors
The authors synthesize a new azo prodrug (CAA) linking carbutamide with salicylic acid to achieve colon-specific delivery of CBT and 5-ASA. They characterize its physicochemical properties, assess metabolic stability and colonic activation, and demonstrate in vivo efficacy in a DNBS-induced rat colitis model. CAA shows superior anti-inflammatory effects compared with sulfasalazine (SSZ), enhanced synergy between CBT and 5-ASA, and reduced risk of systemic hypoglycemia compared to direct CBT administration .
That said, I do have some concerns:
Translational Relevance: While the rat DNBS-colitis model is appropriate, human relevance remains uncertain. More discussion on interspecies differences in colonic azoreductase activity and PepT1 expression is needed to justify clinical translation.
Mechanistic Depth: The proposed dual mechanism (anti-inflammatory and PepT1 inhibition) is promising but only partially substantiated. The PepT1 inhibition assay (Fig. 5) could be complemented with molecular or immunohistochemical confirmation of PepT1 modulation in colonic tissue.
Safety Profile: The manuscript mentions bone marrow toxicity of CBT in historical reports . While hypoglycemia risk is addressed, bone marrow suppression and other systemic toxicities should be discussed in more detail, including future preclinical safety plans.
Comparative Benchmarks: The study benchmarks CAA only against SSZ. Since olsalazine and balsalazide are also clinically relevant azo-prodrugs of 5-ASA, including a brief discussion or rationale for excluding them from experimental comparison would strengthen the manuscript.
I also have some minor comments, which I would recommend addressing:
Language/Clarity: Generally well-written, but several sections could be streamlined for conciseness (e.g., Methods are lengthy; consider moving some details to Supplementary Information).
Figures: Figures 2–4 include error bars but should specify whether values represent SD or SEM in the legends.
Histological images (H&E, p. 11 and 13) would benefit from scale bars.
References: Ensure consistency in citation formatting (some references lack spaces before brackets).
Abbreviations: The list on p. 17 is useful, but “CAA” should be consistently defined early and repeated at least once in the Discussion.
Reviewer 3 Report
Comments and Suggestions for Authors
A manuscript on drug repurposing for carbutamide, via azo-produg synthesis is presented. The research is novel and I believe it would be very interesting for journal's readers. However, this reviewer thinks some points merit clarification previous acceptance:
Major issues:
1) Permeability assays do not show permeability results [cm/s], but mass of compound in the receiver compartment (dM/dt). Please rename the section, otherwise, a permeability result is expected. For this, sac surface must be known, otherwise Perm. coefficients cannot be calculated.
1.1) Furthermore, experimental controls in this assay may be improved. For instance, mass balance control are usually included in transport assays, as well as barrier integrity controls using low permeability markers. How can one know that high flux was not caused by any toxic effect on tissue barrier integrity?
1.2) Analytics in this assay also need clarification. CBT was quantified via UV-spectrophotometry at lambda max= 270 nm, though many aminoacids and peptides absorb at 280 nm. Considering the media is in intimate contact with tissue sacs, how can UV provide reliable measurements of CBT? How was the selectivity issue of this method prevented/accounted for?
1.3) Said potential analytical problem may be the explanation for some inconsistencies shown in the results: Fig. 2 shows that a plateau (equilibrium) seems to be reached after CBT transport in both treatments. However, the mass transported corresponds to no more than 1% of the mass in the donor solution. This may not be expected for such a hydrophilic compound (considering log D values provided in the paper).
1.4) Similar questions can be asked about results presented in Fig 5. VCV transport seems to be improved by combination of 5-asa and CBT. A permeability marker or barrier integrity control is needed to avoid any misinterpretation of toxic effects mediating the enhanced flux.
2) On the mentioned Fig 5. how do the authors explain the cooperative Pept-1 inhibition of CBT and 5ASA? Shouldn't they also inhibit each other? Is there any evidence of synergestic effect between them? Considering that VCV is substrate of other transporters, could this inhibition be the consequence of CBT and 5ASA targeting more than one transporter?
3) Why was CBT withdrawn from the market? This is not explicitly mentioned in the introduction and only line 589 and 590 in discussion bring some hints on this. This is something that must be explicit in the introduction as it does prevent overselling the article. This comment applies regardless of the fact that it is beyond the scope of the manuscript and further research may be needed.
Minor points:
4) In line 468, Pept1 expression in humans and rodents is mentioned. However, the authors may want to consider not only RNA expression, but also protein abundance. is there any available literature on this?
5)Fig 1 B says microbial enzyme metabolism mediates CAA reduction. However, this reactions may take place without microbial enzymes in reductive environments. Should that be considered?
6) Fig 5 does not show a curve for CAA (non-cleaved) inhibition on VCV kinetics. Should that be relevant too?
7) Titles in methods may be more informative. For instance, section 2.4 may be split in two different methods. Section 2.6 may be changed by stability?or release?. 2.8 may also be changed to PK? Please reconsider these titles.
Round 2
Reviewer 2 Report
Comments and Suggestions for Authors
Thank you for addressing the comments.
Reviewer 3 Report
Comments and Suggestions for Authors
This reviewer thanks the authors for addressing the comments so thoroughly.